A transformer-based deep learning framework to predict employee attrition

Li Wenhui m15554201711_1@163.com
School of Information Science and Engineering, Shandong Normal University , Shandong , China
Nguyen Binh
Electronic publication date: 2023 Sep 27
Publication date: 2023
Volume: 9
Electronic Location ID: e1570
Received 2023 Jun 7; Accepted 2023 Aug 14
Copyright: ©2023 Li
Copyright year: 2023
Copyright holder: Li
License: This is an open access article distributed under the terms of the Creative Commons Attribution License, which permits unrestricted use, distribution, reproduction and adaptation in any medium and for any purpose provided that it is properly attributed. For attribution, the original author(s), title, publication source (PeerJ Computer Science) and either DOI or URL of the article must be cited.
License URL: https://creativecommons.org/licenses/by/4.0/

Keywords: Data science, Machine learning, Artificial intelligence, Attrition prediction, Deep learning

Funding: The author received no funding for this work.

==============================
In all areas of business, employee attrition has a detrimental impact on the accuracy of profit management. With modern advanced computing technology, it is possible to construct a model for predicting employee attrition to minimize business owners’ costs. Despite the reality that these types of models have never been evaluated under real-world conditions, several implementations were developed and applied to the IBM HR Employee Attrition dataset to evaluate how these models may be incorporated into a decision support system and their effect on strategic decisions. In this study, a Transformer-based neural network was implemented and was characterized by contextual embeddings adapting to tubular data as a computational technique for determining employee turnover. Experimental outcomes showed that this model had significantly improved prediction efficiency compared to other state-of-the-art models. In addition, this study pointed out that deep learning, in general, and Transformer-based networks, in particular, are promising for dealing with tabular and unbalanced data.

Introduction

Employee attrition (or staff turnover) refers to a natural process by which employees leave their companies for a variety of reasons. There are diverse factors that might lead to an employee’s resignation (Peng, 2022). The hiring procedure is difficult and time-consuming for the majority of companies, yet employees can quickly decide to leave their jobs. The companies suffer workload crises with vacancies when employees leave. The employee attrition rate is a crucial indicator providing valuable insights into the level of business development. The high attrition rate means that workers regularly quit their jobs (Raza et al., 2022). However, the high rate can damage the company’s structure and result in management loss. To secure normal operation, company managers, therefore, have to effectively control their employee attrition rates. To fully understand the attrition process, different types of employee attrition need to be investigated (Li et al., 2023). ‘Internal attrition’ occurs when an employee gets elevated to a higher level within the same firm while ‘external attrition’ happens once an employee quits their present job to work for another company. ‘Involuntary attrition’ happens when a company terminates its employee’s contract. On the contrary, the ‘voluntary attrition’ is determined when an employee decides to leave their company. To calculate the attrition rate, the count of ex-workers is divided by the average count of current workers in a particular period and this rate allows company managers to assess the company’s efficiency over that period. Statistically, one-third of new workers have tendencies to quit their jobs after six months (Peng, 2022). According to the Job Openings and Labor Turnover Survey (JOLTS), up to 4.5 million employees leave their jobs per month in the United States (Raza et al., 2022). The Apollo Technical study indicated that the employee attrition rate was about 19% in many industries (Raza et al., 2022). While in America, the Bureau of Labor Statistics reported that the employee attrition rate reached over 57.0% in 2021 (Raza et al., 2022). For optimal maintenance of the company’s operation, the employee retention rate is desired to be at 90% while the attrition rate is expected to be below 10%.

The age of artificial intelligence technologies and information explosion has triggered the growth of machine learning in which machines are trained with known data (Long et al., 2023; Huang et al., 2021) and then are used to predict future outcomes (Gandomi, Chen & Abualigah, 2022; Xiao et al., 2023; Liu et al., 2020). Machine learning is essential in the field of data science. In a variety of organizations, machine learning is utilized for decision-making to provide more accuracy and labor-saving. Qutub et al. (2021) proposed an automated framework to predict employee attrition by using a series of models constructed with several conventional machine learning algorithms and data collected from the IBM HR employee database. Habous, Nfaoui & Oubenaalla (2021) utilized AdaBoost, Decision Tree, Random Forest, Gradient Boosting, and Logistic Regression to develop the prediction models for employee attrition. Their Logistic Regression model achieved an accuracy of 86%. Najafi-Zangeneh et al. (2021) proposed a Logistic Regression model incorporated with the max-out feature selection technique for dimension reduction; however, their model achieved an accuracy of 81% only. Pratt, Boudhane & Cakula (2021) conducted a comparative assessment among various models to find cutting-edge machine learning approaches for the prediction of attrition rate. Sadana & Munnuru (2022) used machine learning to investigate the attrition problems in IT companies and their findings alerted the company’s management board to adjust working conditions. Being aware of this information enabled the companies to quickly take action to retain the workers who had been dissatisfied with their work, office atmosphere, work-life balance, promotion opportunities, and other factors. Kaya & Korkmaz (2021) constructed a variety of machine learning models in combination with feature selection, class rebalancing, and bootstrapping techniques. The advent of these approaches helps improve employee satisfaction by accurately predicting the staff turnover rate. Generally, existing approaches mainly focus on applying conventional machine learning algorithms and boosting the performance of the model with several techniques to address class imbalance issues. Although these methods showed satisfactory findings, exploration of more advanced and effective methods is highly essential to deal with greater volumes of data.

In this study, we propose a more robust computational framework using a Transformer-based neural network designed for tubular data (Huang et al., 2020) to solve the problem of employee attrition. For model training and evaluation, we used the IBM HR Analytics Employee Attrition dataset (Aizemberg, 2019). Since most previous approaches employed conventional machine learning for modeling strategies, we compare our model to a list of state-of-the-art models which are constructed with conventional machine learning algorithms, including Gradient Boosting (Friedman, 2001; Friedman, 2002), Extreme Gradient Boosting (Chen & Guestrin, 2016), Random Forest (Breiman, 2001), and Extremely Randomized Trees (Geurts, Ernst & Wehenkel, 2006) to fairly assess the improved efficiency. All algorithms selected for benchmarking are tree-based algorithms, which are widely used to deal with class imbalance issues.

Materials and Methods

Overviews

Figure 1 summarizes the major steps in our study. First, the collected dataset was divided into two parts: training data (85%) and test data (15%). While the training data are then used for model optimization and development, test data are kept separately to avoid any data leakage. To train deep learning models, 15% of total training data are used to create a validation set, and the rest of the data are used for model development. The validation set plays a role in finding the optimal models. To train the machine learning model, the training data are used without creating an independent validation set. A 5-fold cross-validation is applied to the training data to compute the average performance corresponding to specific sets of parameters to find the best hyperparameters. The machine learning models are then retrained with training data and hyperparameters. Finally, the deep learning models and machine learning models are benchmarked using the independent test set.

Figure 1 Overviews of the study.

Data preparation

IBM Analytics provides the IBM HR Analytics Employee Attrition dataset (Aizemberg, 2019) which was used in this study. The dataset contains information about various factors that can contribute to employee attrition, including demographic information, job satisfaction, job involvement, performance ratings, and other factors. The dataset had 1,470 samples with 16.1% (237 samples) who left their companies while 83.9% (1,233 samples) kept their job position. The dataset has 35 variables with 26 continuous and nine categorical variables. Stratified random sampling was used to generate the development and test sets, with percentages of 85% and 15%, respectively. In addition, the sampling method also was used to produce the validation set from 10% of the development data. The validation set was assigned to supervise the training process to find the optimal model. independent test set and the development set were checked to assure that no data duplicates were found in both sets. After utilizing one-hot encoding on categorical variables, normalization was employed to ensure that all variables are on a common scale and eliminates the influence of large values on the model’s predicted outcomes. The data can be used to build predictive models to identify employees who are at risk of leaving the company, as well as to develop strategies for improving employee satisfaction and engagement. Table 1 gives information on the distribution of samples of development, validation, and test sets.

Table 1 Data statistics of development, validation, and test sets.

Dataset	Employee attrition status	
	Leaving	Staying	Total	
Development set	181	943	1,124	
Validation set	20	105	125	
Independent test set	36	185	221	

Model architecture

Figure 2 describes the architecture of our proposed method. Overall, the model was constructed with one column embedding layer, one N-stacked Transformer layer, and one multilayer perceptron (Huang et al., 2020). A Transformer block is specified by two components, including a multi-head self-attention layer and a position-wise feed-forward layer. For a feature-target pair (x, y) where x ≡ {xcategorical, xcontinuous}, xcategorical and xcontinuous represent for all categorical features and continuous features. Hence, xcategorical = {x1, x2, x3, …, xm} contain a set of categorical features xi which are independently embedded using the column embedding of dimension d. The embedding of feature xi is denoted as eϕi(xi) ∈ℝd. For all the existing categorical features, the set of embeddings is defined as Eϕ(xcategorical) = {eϕ1(x1), eϕ2(x2), eϕ3(x3), …, eϕm(xm)} which are passed through the first Transformer layer. The first Transformer layer’s output is then fetched to the second Transformer layer as input. The process is successively repeated in the same manner until passing the Nth Transformer layer. Each embedding is finally converted into contextual embedding after exiting the last Transformer layer. Briefly, the series of Transformer layers can be defined as a function fω which map embeddings Eϕ(xcategorical) = {eϕ1(x1), eϕ2(x2), eϕ3(x3), …, eϕm(xm)} to create the corresponding contextual embeddings Hϕ(xcategorical) = {h1, h2, h3, …, hm} ∈ ℝd. The contextual embeddings Hϕ(xcategorical) are then concatenated with the continuous features xcontinuous to generate a new vector of dimension (d + m + c). This vector is operated by the multilayer perceptron layer (denoted as gϵ) afterward to return the predicted probability for target y. The loss function (denoted as K) for model training is expressed as: (1) Lx,y≡KgϵfωEϕxcategorical,xcontinuous,y.

Figure 2 Model architecture for prediction of employee attrition.

Transformer block is specified by one Multi-head Attention layer, one Linear layer, and two shortcut connections. The output of the Transformer block and that of the Normalization layer are then concatenated before passing through multiple Linear layers.

Column embedding

For a column of categorical feature i, a lookup table for embedding eϕi(.) is defined. If a categorical feature xi has di classes, the lookup table eϕi(.) is specified by (di + 1) embeddings in which the additional embedding corresponds to a missing value. The feature xi = j ∈ [0, 1, 2, 3, …, di] is encoded with a new embedding eϕi(j) = [cϕj, wϕi,j] where cϕj ∈ ℝl and wϕi,j ∈ ℝd−l with the dimension cϕj and l are hyperparameters for tuning. The distinct identifier cϕj ∈ ℝl differentiates the classes from one column to other columns. Positional encodings are not employed for tabular data since there is no feature order available, in contrast to language models where the embeddings are inserted element by element with the positional encoding of the word in the sentence.

Modeling experiments

The Adam optimizer (Kingma & Ba, 2014) was used to repeatedly adjust network parameters at a learning rate of 5 × 10−5. During the training process, the network was monitored to find its optimal state when its validation loss bottomed. After 80 training epochs, we found that our model’s validation loss converged at epoch 27th. Hence, we selected the model at epoch 27th as our optimal model. Since the problem is binary classification, we chose binary cross-entropy as our loss function: (2) Loss= ∑i=1nyi× logy ^i+1−yi× log1−y ^i,

where y is the actual label and y ^ is the predicted probability. In our modeling experiments, we designed and tested all deep learning models under the PyTorch 1.3.1 platform. These models were trained on an Intel i7-12700 CPU with 64GB RAM and an NVIDIA GeForce RTX 3090 Ti GPU. One epoch required around 1.2 s for training and 0.2 s for testing.

Assessment metrics

In this study, a number of metrics were used to evaluate the performance of the models, including the area under the receiver operating characteristic curve (AUCROC), the area under the precision–recall curve (AUCPR), and accuracy (ACC). These metrics were calculated based on the True Positive, False Positive, True Negative, and False Negative values. The AUCROC and AUCPR are significant metrics in machine learning for evaluating classification models. The AUCROC measures the model’s ability to accurately classify positive and negative instances, while the AUCPR focuses on precision and recall trade-offs, particularly in imbalanced datasets. Both metrics provide comprehensive assessments of the model’s performance, are robust to class imbalance, and simplify the evaluation process. They help researchers and practitioners make informed decisions about model selection and optimization, providing valuable insights into the classifier’s discriminative ability.

Results and Discussion

Model assessment

To train the model, stratified random sampling was used to select 10% of the original development set to construct a validation set to monitor the training, while the remaining data were used for training. The models’ validation loss converged around epoch 27th. Both training and validation loss continues to slightly decrease after 20 epochs. The best model was chosen at which epochs resulted in the lowest validation loss. To enhance the performance of the model, the optimal model was reloaded and then trained with one, two, and three additional epochs using the whole original development set. The learning rate was set at 1 × 10−5. For comparison, the following three models were obtained: (a) the model trained with one additional epoch, (b) the model trained with two additional epochs, and (c) the model trained with three additional epochs. The model of setup (a) shows better performance than the models of setups (b), and (c). The variations in AUCROC values, however, are not significantly different. Models of setup (a), (b), and (c) have AUCPR values of around 0.38. The AUCROC value of 0.75 is consequently incredibly meaningful for addressing the problem of employee attrition prediction (Table 2).

Table 2 Performance of our models with different setups and state-of-the-art models on the test set. Models (a), (b), and (c) were trained with 1, 2, and 3 additional epochs, respectively.

Bold indicates the highest value corresponding to each metric.

Model	AUCROC	AUCPR	ACC	
ERT	0.7009	0.3717	0.8462	
GB	0.6929	0.3590	0.8371	
RF	0.6812	0.3437	0.8326	
XGB	0.6542	0.2875	0.8281	
Our model (a)	0.7452	0.3849	0.8507	
Our model (b)	0.7434	0.3823	0.8507	
Our model (c)	0.7432	0.3825	0.8507	

Comparative analysis

We also trained several state-of-the-art models using conventional machine learning algorithms to predict employee attrition. The computational frameworks were constructed with Gradient Boosting (Friedman, 2001; Friedman, 2002), Extreme Gradient Boosting (Chen & Guestrin, 2016), Random Forest (Breiman, 2001), and Extremely Randomized Trees (Geurts, Ernst & Wehenkel, 2006) which are abbreviated as GB, XGB, RF, and ERT, respectively. All models were tuned with selected parameters via the grid search method to get the best model performance. We performed grid searches with 5-fold cross-validation on the whole original development set to tune the GB, XGB, RF, and ERT models. For the deep learning model, the validation set (split from the original development set) was used separately to determine the optimal epoch in which the model exhibits the smallest validation loss. The comparative analysis provides evidence that our proposed method can work better than other conventional machine learning models which have been commonly used to address these types of problems (Table 2). In addition, our research suggests constructing more effective prediction tools by using advances in deep learning. Since there is no significant difference in performances found among these setups (a), (b), and (c), we selected the model of setup (a) as our official model to conduct a comparative analysis with the other state-of-the-art approaches. Achieving a test AUCROC value of about 0.75, our model demonstrates a substantial improvement in predictive efficiency over other models. For an imbalanced-class dataset, the AUCPR value is more valuable than the accuracy when evaluating a binary classification model. Our models yielded an AUCPR value of 0.38, whereas other methods obtain AUCPR values of at most 0.37. Figure 3 visualizes the areas under the curves of all the models.

Figure 3 Areas under the curves of all the models.

A. Receiver operating characteristic curves, B. Precision–recall curves.

Statistical analysis

To investigate the robustness of the models, we replicated the experiments with different random seeds to avoid sampling bias. Table 3 gives information on the performance of all models over ten trials. The results indicated that our model outperforms the other machine learning models with AUCROC and AUCPR values of 0.7632 and 0.4792, respectively. In terms of AUCROC, the ERT and GB models show fairly equivalent performance, followed by the RF and XGB models. While the ERT model is still the best conventional approach compared to the others, the RF model achieves higher AUCPR values than the GB and XGB models. Also, to assess the statistical significance of the results, we used two-tailed independent t-tests with a confidence interval of 0.95 to compare the performance of our model to that of each machine learning model (Table 4). The p-values of these pairwise comparisons between our model and the other models confirm the statistical significance of these results.

Table 3 The performance of repetitive runnings.

Metric	Model	
	ERT	GB	RF	XGB	Ours	
AUCROC	0.6778	0.6922	0.6793	0.6542	0.7387	
0.7047	0.6844	0.6817	0.6498	0.7202	
0.688	0.6934	0.6851	0.6661	0.7579	
0.6898	0.6887	0.6977	0.6474	0.7431	
0.6988	0.6869	0.6847	0.695	0.8333	
0.7047	0.6989	0.6818	0.695	0.7611	
0.6821	0.7024	0.7081	0.695	0.8089	
0.7086	0.6941	0.7032	0.695	0.7587	
0.6865	0.6822	0.6794	0.695	0.7609	
0.6935	0.6847	0.6965	0.6542	0.7490	
MeanAUCROC	0.6934	0.6908	0.6898	0.6747	0.7632	
SDAUCROC	0.0104	0.0066	0.0106	0.0220	0.0335	
AUCPR	0.3509	0.3568	0.3101	0.2875	0.5514	
0.3723	0.3467	0.3191	0.2907	0.4111	
0.3433	0.3502	0.3578	0.3087	0.4758	
0.3650	0.3382	0.3729	0.2953	0.4278	
0.3622	0.3502	0.3648	0.3212	0.5422	
0.3676	0.3453	0.3548	0.3212	0.4139	
0.3573	0.3592	0.3523	0.3212	0.5156	
0.3702	0.3334	0.3740	0.3212	0.3833	
0.3537	0.3505	0.3570	0.3212	0.5433	
0.3715	0.3383	0.3413	0.2875	0.5275	
MeanAUCPR	0.3614	0.3469	0.3504	0.3076	0.4792	
SDAUCPR	0.0098	0.0083	0.0213	0.0155	0.0647	

Table 4 Pairwise independent t-test comparing machine learning models against ours.

P-value on metric	Model	
	ERT	GB	RF	XGB	
AUCROC	6.36914 × 10−6*	2.7768 × 10−6*	3.3631 × 10−6*	1.6023 × 10−6*	
AUCPR	2.6357 × 10−18*	1.1465 × 10−18*	1.5935 × 10−17*	7.6555 × 10−19*	
Notes.

* indicates statistical significance.

Discussion

As a transformer-based model (or, shortly, Transformer), our work also has limitations that need to be improved in the future. Generally, although Transformer-based models achieve great success in various natural language processing tasks, they have limitations when working with small datasets. With limited training samples, Transformer-based models are prone to overfitting and memorizing data instead of learning patterns. The lack of diversity in small datasets can hinder the model’s ability to generalize and handle unseen data effectively. Furthermore, the insufficient contextual information in small datasets makes it challenging for Transformer-based models to grasp the semantics and relationships within the data, leading to suboptimal performance. Data sparsity is also a concern, as infrequent or absent words and patterns can impede the learning process. Finally, the high capacity of Transformer-based models may be underutilized with small datasets, limiting their ability to capture complex relationships. Mitigation strategies include transfer learning, data augmentation, regularization techniques, and domain adaptation. These approaches can partially address the limitations, but it is important to acknowledge the inherent challenges of training large-scale models with small datasets. Transfer learning allows leveraging knowledge from related tasks or domains, while data augmentation increases training data diversity. Regularization techniques prevent overfitting and improve generalization, and domain adaptation aligns representations for better adaptation to new domains. These strategies enhance the Transformer-based model’s performance, generalization, and adaptability.

Limitations

Despite good outcomes, our model still has limitations that need to be improved in the future. Like other models in the Transformer family, our model requires high computational cost compared to other deep learning architectures. Besides, longer training duration and limited parallelization are also common issues of Transformer-based models. On the other hand, parameter tuning in a Transformer-based model is highly sensitive to create the optimal models.

Conclusions

Machine learning techniques have shown promise in predicting employee attrition by leveraging large and diverse datasets to uncover hidden patterns and complex relationships. Accurate attrition predictions can help organizations take proactive measures to retain valuable employees and maintain a stable workforce. However, addressing challenges related to data quality, privacy, interpretability, and ethics is crucial to ensuring the effective and responsible use of machine learning in employee attrition prediction. Under the scope of this study, the experimental results indicated that our proposed method is an effective computational framework to predict employee attrition. Besides, our method obtained higher performance than other state-of-the-art methods. In addition, it is highly advised that deep learning can be a promising modeling option to deal with tubular data with imbalanced classes besides frequently used conventional machine learning. In the future, our approach will be developed to be more applicable to a wider variety of issues with similar data types.

Supplemental Information

Supplemental Information 1 Source code and data

Click here for additional data file.

Additional Information and Declarations

Competing Interests

Author Contributions

Data Availability

The author declares that they have no competing interests.

Wenhui Li conceived and designed the experiments, performed the experiments, analyzed the data, performed the computation work, prepared figures and/or tables, authored or reviewed drafts of the article, and approved the final draft.

The following information was supplied regarding data availability:

The data and code are available in the Supplemental File.

The data used in this study is from Aizemberg (2019) and available at https://data.world/aaizemberg/hr-employee-attrition.

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
