# Peer review of "A transformer-based deep learning framework to predict employee attrition"

_PeerJ Computer Science, doi:10.7717/peerj-cs.1570_

## Round 0.1 · original submission · Major Revisions

Please address the comments from the two reviewers, especially the comments about the stability of the model (Reviewer 1) and conducting statistical testing (Reviewer 2), and then revise the manuscript accordingly.

Reviewer 1 ·

Basic reporting

This article discusses utilising advanced computing technology to predict employee attrition and manage business profits more accurately. It presents a study that applied a Transformer-based neural network to the IBM HR Employee Attrition dataset, resulting in improved prediction efficiency. The study also underscores the promise of deep learning for handling unbalanced tabular data.

The document is well-structured and exhibits proficient use of language. While the background information is generally adequate, it would be better if the author could update some most recent studies that applied this model. The presentation, such as figures and tables, is clearly displayed; however, the author should give more details in the figure legend to make it self-explained.

Experimental design

The study is in line with the journal's Aims and Scope and contains concise research questions. The model's structure is thoroughly described, and the provided methodologies are sufficiently informative. Still, the "Assessment Metrics" parts need additional explanations of the significance of these metrics.

Validity of the findings

Model robustness and stability are still questionable. Conducting experiments using different data sampling trials is highly recommended to have more statistical evidence to support the conclusion of the study.

Additional comments

Besides, there are still a few significant issues that need to be addressed:
- What is the significance of the metrics mentioned in the Assessment Metrics? Also, why did you choose these metrics for assessment?
- Table 2: What is ERT, GB, RF, and XGB? Please include the information for these abbreviations in the introduction and the Table legend.
- Line 175: ‘Mitigation strategies include transfer learning, data augmentation, 175 regularization techniques, and domain adaptation.” Why can these strategies address the limitation of the Transformer? I suggest having a different paragraph for this information for more detailed information.

Cite this review as

Reviewer 2 ·

Basic reporting

The manuscript is well-organized and follows the official English writing style. The literature review says enough about the subject to give a full idea of it. The study results are presented in a clear and concise way in the manuscript by using well-designed tables and figures. Still, there are some things that need to be fixed before the story is ready to be published. The list below shows specific questions that need to be answered.

Experimental design

The experimental design appears to be sound; nevertheless, there are certain areas that require clarification. Specifically, it is important to ascertain which datasets were utilized for training and tuning the GB, XGB, RF, and ERT models. Although the manuscript mentions the utilization of grid search, it remains unclear whether it was employed in conjunction with cross-validation or solely based on the validation set. Furthermore, considering the imbalanced nature of the dataset, it would be beneficial for the authors to explore the implementation of data balancing techniques during the development of the models.

Validity of the findings

The findings of the study are useful for researchers across diverse fields with an interest in related subjects. The provision of a publicly available dataset and code repository facilitates result replication and further exploration of the study. However, as the dataset is relatively small, it raises concerns about potential bias in data splitting. Consequently, it becomes challenging to ascertain whether the high performance stems from a robust model or biased random sampling. To address this, it is necessary to conduct statistical testing, such as a t-test, to compare the proposed model with other baseline models. This would provide a more comprehensive evaluation and enhance the validity of the study's claims.

Cite this review as

---

## Round 0.2 · Minor Revisions

Please revise the manuscript to address the comments from the two reviewers.

Reviewer 1 ·

Basic reporting

The revised manuscript has undergone significant improvements, with a stronger emphasis on scientific evidence to support the study's motivation. The authors addressed all of the major concerns raised about their work, resulting in a more comprehensible version to readers. I suppose that the current version is now suitable for publications with minor modifications:
- Line 98: “.... For a feature-target pair (x, y) where x ...”: “(x, y)” and “x” need to be italicised like other mathematical signs, variables, and operators.
- Line 128: “... In our modeling experiment ...” -> In our modeling “experiments” (plural)
- Line 130-131: “... One epoch required around 1.2 seconds to train and 0.2 seconds for testing.” -> Your sentence should written in a parallel structure. “to train” -> “for training”.
- Line 147: ... epoch 27. -> “... epoch 27th”
- Line 153-154: “... than the other setup models (b), and (c).” -> “... than the models of setups (b) and (c).”
- Line 175: “... value of 0.38, whereas other methods obtains an AUCPR value” -> value of 0.38, whereas other methods “obtain” (fixed verb) AUCPR “values” (plural)
- Line 178-179: “Table 3 gives information on the performance of all models over multiple trials.” -> “Table 3 gives information on the performance of all models over “ten” (concrete number) trials.
- Line 186: “The p-values” -> The p-values (italicised “p”)
- Line 190, 192, 197, 204: “Transformers” -> “Transformer-based models”

Experimental design

The experiments were well-designed to achieve the study’s objectives.

Validity of the findings

The newly added statistics provide more insights into the model's robustness and applicability.

Additional comments

I have no additional comments for this article.

Cite this review as

Reviewer 2 ·

Basic reporting

I appreciate the authors’ efforts in adding more details and conducting additional experiments to improve the quality of the manuscript. The manuscript is now well-structured and understandable to readers. I recommend that this work be considered for publication once the authors have completed correcting several minor points.
(1) Figure 2. It is recommended to move the legend of Figure B to the top-right position to avoid overlaying text.
(2) Figure 2. The axis names in Figure B are wrong. It should be "Precision" and "Recall" instead of "TPR" and "FPR".
(3) The limitations of the method should be discussed.
(4) In the statistical analysis section, which threshold did you choose? (0.05, 0.01, etc.). Is it the one-tail or two-tail test?

Experimental design

The description of the experiments is clear and simple. It would be much better if the authors could add a flowchart describing the major steps in conducting their experiments.

Validity of the findings

The work provides sufficient results in terms of experiments and statistical testing to evaluate the validity of the findings. Based on their results, I agree that their proposed method works better than other methods.

Additional comments

- Line 75: "All selected algorithms for" should be read "All algorithms selected for".
- Line 185: "compare the performance of our model to each machine learning model" should be read "compare the performance of our model to that of each machine learning model".

Cite this review as

---

## Round 0.3 · accepted · Accept

The authors have made efforts to address all of the reviewers' comments. I have assessed the revised manuscript and acknowledge that the authors have adequately incorporated the feedback. It is not necessary to send this version to the reviewers. Based on this assessment, the manuscript is now ready for publication.

---

## Author Rebuttal · Round 0.3

**Response to Reviewer 1**

**Comment**:

Basic reporting

The revised manuscript has undergone significant improvements, with a stronger emphasis on scientific evidence to support the study's motivation. The authors addressed all of the major concerns raised about their work, resulting in a more comprehensible version to readers. I suppose that the current version is now suitable for publications with minor modifications:

**Response**:

Thank you for your supportive comments. We have significantly revised our manuscript to correct all the mentioned errors.

**Comment**:

- Line 98: ".... For a feature-target pair (x, y) where x ...": "(x, y)" and "x" need to beitalicised like other mathematical signs, variables, and operators.

**Response**:

109    attention layer and a position-wise feed-forward layer. For a feature-target pair $(x, y)$ where $x \equiv$
110    $\{x_{categorical}, x_{continuous}\}$, $x_{categorical}$ and $x_{continuous}$ represent for all categorical features and continuous

**Comment**:

- Line 128: "... In our modeling experiment ..." -> In our modeling "experiments"(plural)

**Response**:

139    where $y$ is the actual label and $\hat{y}$ is the predicted probability. In our modeling experiments, we designed

**Comment**:

- Line 130-131: "... One epoch required around 1.2 seconds to train and 0.2 secondsfor testing." -> Your sentence should written in a parallel structure. "to train" -> "for training".

**Response**:

141    Intel i7-12700 CPU with 64GB RAM and an NVIDIA GeForce RTX 3090 Ti GPU. One epoch required
142    around 1.2 seconds for training and 0.2 seconds for testing.

**Comment**:

- Line 147: ... epoch 27. -> "... epoch 27th"

**Response**:

158  The models' validation loss converged around epoch $27^{th}$. Both training and validation loss continues

**Comment**:

- Line 153-154: "... than the other setup models (b), and (c)." -> "... than the models of setups (b) and (c)."

**Response**:

164  with three additional epochs. The model of setup ($a$) shows better performance than the models of setups
165  ($b$), and (c). The variations in AUCROC values, however, are not significantly different. Models of

**Comment**:

- Line 175: "... value of 0.38, whereas other methods obtains an AUCPR value" ->value of 0.38, whereas other methods "obtain" (fixed verb) AUCPR "values" (plural)

**Response**:

186  value of 0.38, whereas other methods obtain AUCPR values of at most 0.37. Figure 3 visualizes the areas
187  under the curves of all the models.

**Comment**:

- Line 178-179: "Table 3 gives information on the performance of all models over multiple trials." -> "Table 3 gives information on the performance of all models over "ten" (concrete number) trials.

**Response**:

190  to avoid sampling bias. Table 3 gives information on the performance of all models over ten trials. The

**Comment**:

- Line 186: "The p-values" -> The p-values (italicised "p")

**Response**:

197  each machine learning model (Table 4). The $p$-values of these pairwise comparisons between our model
198  and the other models confirm the statistical significance of these results.

**Comment**:

- Line 190, 192, 197, 204: "Transformers" -> "Transformer-based models"

**Response**:

206 makes it challenging for Transformer-based models to grasp the semantics and relationships within
207 the data, leading to suboptimal performance. Data sparsity is also a concern, as infrequent or absent
208 words and patterns can impede the learning process. Finally, the high capacity of Transformer-based
209 models may be underutilized with small datasets, limiting their ability to capture complex relationships.
210 Mitigation strategies include transfer learning, data augmentation, regularization techniques, and domain
211 adaptation. These approaches can partially address the limitations, but it is important to acknowledge the
212 inherent challenges of training large-scale models with small datasets. Transfer learning allows leveraging
213 knowledge from related tasks or domains, while data augmentation increases training data diversity.
214 Regularization techniques prevent overfitting and improve generalization, and domain adaptation aligns
215 representations for better adaptation to new domains. These strategies enhance the Transformer-based
216 model's performance, generalization, and adaptability.

217 **Limitations**
218 Despite good outcomes, our model still has limitations that need to be improved in the future. Like other
219 models in the Transformer family, our model requires high computational cost compared to other deep
220 learning architectures. Besides, longer training duration and limited parallelization are also common
221 issues of Transformer-based models. On the other hand, parameter tuning in a Transformer-based model
222 is highly sensitive to create the optimal models.

**Comment**:

**Experimental design**

The experiments were well-designed to achieve the study's objectives.

**Validity of the findings**

The newly added statistics provide more insights into the model's robustness and applicability.

**Additional comments**

I have no additional comments for this article.

**Response**:

Thank you for your supportive comments.

**Response to Reviewer 2**

**Comment**:

**Basic reporting**

I appreciate the authors' efforts in adding more details and conducting additional experiments to improve the quality of the manuscript. The manuscript is now well structured and understandable to readers. I recommend that this work be considered for publication once the authors have completed correcting several minor points.

**Response**:

Thank you for your supportive comments. We have significantly revised our manuscript to correct all the mentioned errors.

**Comment**:

(1) Figure 2. It is recommended to move the legend of Figure B to the top-right position to avoid overlaying text.

(2) Figure 2. The axis names in Figure B are wrong. It should be "Precision" and"Recall" instead of "TPR" and "FPR".

**Response**:

[Figure]

**Figure 3.** Areas under the curves of all the models (A. Receiver operating characteristic curves, B. Precision-recall curves).

**Comment**:

(3) The limitations of the method should be discussed.

**Response**:

 **Limitations**

 Despite good outcomes, our model still has limitations that need to be improved in the future. Like other
 models in the Transformer family, our model requires high computational cost compared to other deep
 learning architectures. Besides, longer training duration and limited parallelization are also common
 issues of Transformer-based models. On the other hand, parameter tuning in a Transformer-based model
 is highly sensitive to create the optimal models.

**Comment**:

(4) In the statistical analysis section, which threshold did you choose? (0.05, 0.01,etc.). Is it the one-tail or two-tail test?

**Response**:

195 the GB and XGB models. Also, to assess the statistical significance of the results, we used two-tailed
196 independent $t$-tests with a confidence interval of 0.95 to compare the performance of our model to that of
197 each machine learning model (Table 4). The $p$-values of these pairwise comparisons between our model

**Comment**:

**Additional comments**

- Line 75: "All selected algorithms for" should be read "All algorithms selected for".

- Line 185: "compare the performance of our model to each machine learning model" should be read "compare the performance of our model to that of each machine learning model".

**Response**:

We have fixed those errors.